

# Withasomniferol C, a new potential SARS-CoV-2 main protease inhibitor from the *Withania somnifera* plant proposed by *in silico* approaches

Shivananada Kandagalla[1], Hrvoje Rimac[2], Krishnamoorthy Gurushankar[1,3], Jurica Novak[1], Maria Grishina[1] and Vladimir Potemkin[1,†]

[1] Higher Medical & Biological School, Laboratory of Computational Modeling of Drugs, South Ural State University, Chelyabinsk, Chelyabinsk, Russia

[2] Department of Medicinal Chemistry, University of Zagreb Faculty of Pharmacy and Biochemistry, Zagreb, Croatia

[3] Department of Physics, Kalasalingam Academy of Research and Education, Krishnankoil, Tamilnadu, India

[†] Deceased.

## ABSTRACT

Exploring potent herbal medicine candidates is a promising strategy for combating a pandemic in the present global health crisis. In Ayurveda (a traditional medicine system in India), *Withania somnifera* (WS) is one of the most important herbs and it has been used for millennia as Rasayana (a type of juice) for its wide-ranging health benefits. WS phytocompounds display a broad spectrum of biological activities (such as antioxidant, anticancer and antimicrobial) modulate detoxifying enzymes, and enhance immunity. Inspired by the numerous biological actions of WS phytocompounds, the present investigation explored the potential of the WS phytocompounds against the SARS-CoV-2 main protease (3CL$^{pro}$). We selected 11 specific withanolide compounds, such as withaphysalin, withasomniferol, and withafastuosin, through manual literature curation against 3CL$^{pro}$. A molecular similarity analysis showed their similarity with compounds that have an established inhibitory activity against the SARS-CoV-2. *In silico* molecular docking and molecular dynamics simulations elucidated withasomniferol C (WS11) as a potential candidate against SARS-CoV-2 3CL$^{pro}$. Additionally, the present work also presents a new method of validating docking poses using the AlteQ method.

## INTRODUCTION

The coronavirus disease 2019 (COVID-19) caused by the new coronavirus represents a significant global health crisis and has posed unprecedented challenges since December 2019, when the first case was reported in Wuhan, China. Later, the new coronavirus was named SARS-CoV-2 (Severe Acute Respiratory Syndrome Coronavirus 2), as it shares 79.5% genome similarity with the other severe acute respiratory syndrome (SARS) virus (*Zhou et al., 2020*; *Lu et al., 2020*). Still, the number of COVID-19 patients is increasing,

Corresponding author
Shivananada Kandagalla,
kandagallas@susu.ru,
kandagallas@gmail.com

with the main cause of recent outbreaks being the new B.1.1.7 (the Alpha variant) (*GOV.UK, 2021*), B.1.351 (the Beta variant) (*Tegally et al., 2021*), P.1 (the Gamma variant) (*Francisco et al., 2021*), B.1.617.2 (the Delta variant), and B.1.1.529 (the Omicron variant). This, along with the presence of asymptomatic carriers, different modes of transmission, a lack of point-of-care diagnostic facilities, and accessibility of approved antiviral drugs and vaccines are the other important factors which are making the COVID-19 situation hard to manage. Additionally, the socioeconomic variables also influence the virus to spread more easily, making it difficult for poorer countries to access vaccines.

Currently, several antiviral drugs (*e.g.,*, remdesivir, bemcentinib, and lopinavir with ritonavir), immune modulators (*e.g.*, azithromycin, brensocatib, anakinra and canakinumab, interferon beta, convalescent plasma, corticosteroids, sarilumab and tocilizumab) are recommended against COVID-19 (*Connelly, 2020*), and their usage mainly depends on patient's symptoms, while remdesivir is approved by the FDA for treatment of COVID-19 (*Beigel et al., 2020*). Presently, the efficiency of nearly fifteen drugs *e.g.*, lopinavir, ritonavir, nafamostat, camostat, famotidine, umifenovir, nitazoxanide, corticosteroids, tocilizumab, sarilumab, bevacizumab, and fluvoxamine are being tested in clinical trials against the virus (*Shaffer, 2020*). Additionally, nearly 64 vaccine candidates are under investigation against the SARS-CoV-2, with most of them aiming to induce synthesis of neutralizing antibodies against the viral spike protein (S), which would thereby prevent interactions with the human ACE-2 receptor (*Kyriakidis et al., 2021*). Among the investigated vaccines, 19 vaccine candidates have completed the phase 3 of clinical trials, and they are approved for immunization programs in most countries (*Kyriakidis et al., 2021*), but their effectiveness against the new SARS-CoV-2 variants still needs to be investigated. Recent reports show that new variants harbor mutations in the S protein, and as a result, this alters viral interactions and ultimately can lead to resistance to antibodies and interaction inhibitors (*Hoffmann et al., 2021*).

Given the emergence of SARS-CoV-2, several studies related to the development of its inhibitors are focused on the three main druggable targets *i.e.*, the 3C-like proteinase (3CL$^{pro}$), the papain-like protease (PL$^{pro}$), and the spike protein. In this study the focus was on the 3CL$^{pro}$, which is also known as the main protease (M$^{pro}$) and plays a key role during viral replication. As with other betacoronaviruses, the SARS-CoV-2 is a positive-sense RNA virus, which expresses all its proteins as a single polypeptide chain. Both the 3CL$^{pro}$ and the PL$^{pro}$ are involved in cleaving the polypeptide chain into mature proteins (*Song et al., 2019*). Among coronaviruses, the 3CL$^{pro}$ three-dimensional structure and its sequence are highly conserved, with a distinct three-domain fold (*Snijder et al., 2003*). Domains I and II have a chymotrypsin-like structure, and together they form the catalytic region, while the $\alpha$-helical domain III is mainly involved in the dimerization process (*Shi, Wei & Song, 2004*; *Chou et al., 2004*). Previous SARS-CoV studies have shown that the 3CL$^{pro}$ is fully functional only in the dimeric form; the monomer has a reduced enzyme activity towards the substrate proteins (*Shi, Wei & Song, 2004*), with both monomers and dimers being observed in solution (*Fan et al., 2004*). The mutation or truncation of the key residues involved in the dimerization of the C-terminus increases the monomer to dimer ratio, which ultimately decreases the enzymatic activity (*Lin et al., 2008*). The N-finger

(residues 1–7) is not involved in the dimerization (*Zhong et al., 2008*) but is required for the activity of the enzyme. In the dimer, the N-finger of the inactive protomer interacts with domains II and III of the active protomer (*Yang et al., 2003*; *Chen et al., 2005*; *Chen et al., 2006*). Recently, different variants of the SARS-CoV-2 main protease were identified from clinical samples, and their structural variations were explored through *in silico* studies, with the conclusion that mutations in these variants do not alter the active site conformation (*Martin et al., 2020*). Hence, the 3CL$^{pro}$ is a good target for drug discovery. So far, peptidomimetic alpha ketoamide inhibitors, the Michael acceptor N3 inhibitor, carmofur, ebselen, aldehyde-based compounds, lopinavir/ritonavir, the antiplatelet drug dipyridamole, boceprevir, GC-376, calpain inhibitors (II, XII), and GC-373 are reported as the most promising drugs against the SARS-CoV-2 3CL$^{pro}$, and they exert their inhibitory effect by binding to the substrate-binding cleft. However, their clinical outcomes in humans are still not determined (*Mengist et al., 2021*).

For decades, natural products and herbal medicines have been used for combating numerous viral infections, with their favorable efficacy and low toxicity making them a promising resource for drug discovery. An acceptable toxicity of natural products and herbal medicines make them prospective candidates against COVID-19 (*Komolafe et al., 2021*). Recent reviews show the significance of natural products against COVID-19 by examining their randomized controlled trial (RCT) reports in COVID-19 patients (*Feng et al., 2021*; *Di et al., 2021*). In this regard, exploring potential herbal medicine candidates is a promising strategy for combating a pandemic in the present global health crisis. *Withania somnifera* (*Solanaceae*, WS), popularly known as 'Ashwagandha' and 'Indian Ginseng' is a medicinal plant used as a herbal tonic to treat various kinds of diseases, such as cancer, arthritis, asthma, aging, inflammation, and neurological disorders in Indian traditional medicine (*Dar, Hamid & Ahmad, 2015*). Its pharmacological activities are mainly due to the presence of diverse secondary metabolites, such as alkaloids, flavanol glycosides, glycowithanolides, steroidal lactones (withanolides), sterols, and phenolic acids. Recently, a clinical trial has been initiated by the AYUSH ministry of India for the use of WS along with other plants to evaluate its efficacy against COVID-19 (CTRI (Clinical Trial Registry –India), registration number: CTRI/2021/06/034496, date of registration: June 30, 2021) (*Chopra et al., 2021*). In addition, the Indian Government has collaborated with the U.K's London School of Hygiene and Tropical Medicine (LSHTM) to conduct a study on "Ashwagandha" for promoting COVID-19 recovery (*India, 2021*). All these reports clearly show WS significance. Hence, evaluating binding interactions of phytocompounds isolated from WS against SARS-CoV-2 proteins may help to understand its mechanism of action against COVID-19. In this concern, in the present investigation, we selected withanolides with a wide range of therapeutic applications (*e.g.*, antimicrobial, anti-tumour, anti-inflammatory, anti-oxidant, anti-stress) for the analysis against the SARS-CoV-2 3CL$^{pro}$. Initially, all withanolides reported in the WS plant were collected through manual literature search, and their binding efficiency against SARS-CoV-2 3CL$^{pro}$ was evaluated using *in silico* molecular docking and MD simulation studies. The docking results were additionally validated using the complementarity principle implemented via the AlteQ method.

## MATERIALS & METHODS

### Collection of phytocompounds

The collection and identification of withanolides from the *Withania somnifera* plant was manually done through literature curation. The NCBI PubMed database search engine (https://www.ncbi.nlm.nih.gov/pubmed/) was used to collect peer-reviewed research articles. All research articles with the search term "*withania somnifera*" were collected for the analysis. Each article was reviewed separately, and the phytocompounds reported in the WS plant with valid experimental evidence were selected for the analysis. The structure information of the phytocompounds was retrieved from the PubChem database in the SMILES format.

### Molecular fingerprinting

A structure based molecular screening of the collected phytocompounds was performed against compounds with a determined SARS-CoV-2 inhibitory activity. A structure based molecular screening was performed using 2D molecular fingerprints (FP) in RDKit (2020.03.1). In the present study, the Morgan circular FP (*i.e.*, the extended connectivity) (*Rogers & Hahn, 2010*) was used to generate the FP of the phytocompounds (the FP radius was set to four). The molecular similarity exploration was performed using the Tanimoto coefficient (Tc) (*Bajusz et al., 2015*). The Tc similarity score ranges from zero to one, with zero representing the minimum and one representing the maximum similarity.

The generated FPs were compared to the calculated FP of 8702 molecules collected from the ChEMBL database (*Davies et al., 2015*) (release version ChEMBL 27) with the assay ID CHEMBL4495582, and the target ID CHEMBL4523582. The assay ID CHEMBL4495582 corresponds to the SARS-CoV-2 inhibitors and target ID CHEMBL4523582 corresponds to the replicase polyprotein 1ab of the SARS-CoV-2.

### Molecular docking

Detailed investigation of phytocompounds against SARS-CoV-2 3CL$^{pro}$ was performed by molecular docking studies. Before docking, 3D conformations of phytocompounds were generated and optimized from their SMILES notations using RDKit and MMFF94 force field. The 3D structures were then converted into the pdbqt file format using AutoDockTools 4 script prepare_ligand4.py (*Morris et al., 2009*). Similarly, the receptor was prepared as described below using UCSF Chimera 1.14 (*Pettersen et al., 2004*). In the present study, we used the SARS-CoV-2 3CL$^{pro}$ conformation which was generated in our previous study based on the k-means clustering of the unbound 3CL$^{pro}$ during a course of a 900 ns MD simulations (PDB ID: 6LU7) (*Novak et al., 2021a*). The k-means clustering was based on all non-hydrogen backbone atoms, and it gave two different conformations of the SARS-CoV-2 3CL$^{pro}$, the primary A conformation and the secondary B conformation, which were present 86.7% and 13.3% of the simulation time, respectively. The structural differences of these conformations compared to the crystallographic conformation of the SARS-CoV-2 3CL$^{pro}$ (6LU7) are thoroughly discussed in (*Novak et al., 2021a*), while for the present work, the A conformation was chosen, as it is the more prevalent one. The A conformation of the SARS-CoV-2 3CL$^{pro}$ was prepared by adding the Gasteiger charges to

all atoms, followed by merging of the nonpolar hydrogens. Such structure was then saved in the pdbqt file format. Docking was performed using AutoDock Vina (*Trott & Olson, 2010a*) locally on a personal computer with 8 Intel® CoreTM i7-6700K CPU @ 4.00 GHz, 32 GB RAM, and 64-bit Windows 10 Pro operating system. The center of the grid box was set on the Cys145 CA atom (dimensions: $x = 13.3$, $y = 58.2$, $z = 45.4$), with a box size of $20 \times 25 \times 25$ Å. The grid spacing was set to 1 Å and number of modes and exhaustiveness were both set to 100.

The phytocompounds' physicochemical descriptors, ADME properties, and druglike nature were evaluated using the SwissADME server (*Daina, Michielin & Zoete, 2017*).

## Selection of docked poses using the AlteQ method

Validation of the conformations generated by the docking tools can be assessed using different methods (*B-H & Brenk, 2009*), among which the RMSD (root-mean-square deviation) based methods are the most widely used. A docked ligand pose with an RMSD score less than 2 Å, compared to the reference ligand (the crystallographically determined ligand pose), is generally considered as a good pose (*Trott & Olson, 2010b*). However, selection of docked poses based solely on RMSD has several flaws and can lead to misclassification of both the correct and incorrect poses (*Cole et al., 2005*; *Kroemer et al., 2004*). In the current work, we used the SARS-CoV-2 3CL^pro conformation from a molecular dynamics simulation, and hence measuring the correctness of docking poses based on the RMSD with the reference model is not an appropriate method. Besides, the covalently bound peptidomimetic ligand of SARS-CoV-2 3CL^pro (6LU7) was used for the MD simulations and measuring the RMSD between covalently bound peptidomimetic ligand with the non-covalently docked ligand is also not appropriate. Therefore, in the present work distance-based measures (*i.e.*, ligand–receptor contacts) were used to check for the correctness of the docked poses. Many distance-based measures have been developed (*Rueda et al., 2010*; *Hawkins et al., 2008*), and in these methods, the cut-off length for the ligand–receptor contacts has to be defined. The complementarity principle coupled with the AlteQ method is a newly developed method for determining ligand–receptor contacts in which no cut-offs have to be introduced (*Potemkin & Grishina, 2008*); it calculates ligand–receptor contacts based on the electron density overlaps between the ligand and the receptor atoms using the Slater's type atomic contributions (*Potemkin, Grishina & Potemkin, 2017*; *Grishina & Potemkin, 2019*). Recently, we used this method to calculate electron density overlaps in EGFR (*Kandagalla et al., 2021*) and CDK (*Rimac, Grishina & Potemkin, 2020*) receptor–ligand complexes, where we showed that all interactions determined by overlaps of electron clouds follow the complementarity principle, expressed by the following equation:

$$ln\left(\rho_{ligand} \times \rho_{enzyme}\right) = b + a \times RLRE. \tag{1}$$

where $\rho_{ligand}$ represents ligand's contribution to electron density in the $m^{th}$ point of the molecular space, $\rho_{enzyme}$ represents enzyme's contribution to electron density in the same point, and RLRE is defined in a following manner (Eq. 2):

$$RLRE = dist_{enzyme} \times dist_{ligand} \tag{2}$$

where $dist_{ligand}$ ($dist_{enzyme}$) represents the distance between the $m$th point and the ligand's (enzyme's) atom having the highest contribution to $\rho_{ligand}$ ($\rho_{enzyme}$) at that point.

The above complementarity model (Eq. 1) was used for validation of the docking poses. In this regard, the complementarity model was developed from the experimentally solved crystal structure of the SARS-CoV-2 3CL$^{pro}$. A total of 29 experimentally solved SARS-CoV-2 3CL$^{pro}$ crystal structures with non-covalently bound inhibitors were selected from the RCSB Protein Data Bank (https://www.rcsb.org/) (*Berman et al., 2002*) for the analysis (PDB ID's 5R7Y, 5R7Z, 5R80, 5R81, 5R82, 5R83, 5R84, 5RE4, 5RE9, 5REB, 5REH, 5REZ, 5RF1, 5RF2, 5RF3, 5RF6, 5RF7, 5RFE, 5RG1, 5RGH, 5RGI, 5RGK, 5RGU, 5RGV, 5RGW, 5RGX, 5RGY, 5RGZ, 5RH0, 5RH1, 5RH2, 5RH3, 5RH8, 5RHd, 6M2N, 6W79, 7JU7, 7KX5, and 7L5D). On the other hand, the five conformations of each phytocompound showing the lowest binding energy with 3CL$^{pro}$ were collected from docking studies for further analysis. All the collected conformations were prepared using the UCSF Chimera 1.14 (University of California, USA) (*Pettersen et al., 2004*). The chain A conformations with the ligands were retained for the electron density analysis. The 3D maps of the electron density ($\rho$) were calculated for all conformations using the in-house developed quantum free-orbital AlteQ method (*Potemkin & Grishina, 2008*). The AlteQ method represents molecular electron density as a sum of Slater's type atomic increments, and it can be expressed as (Eq. 3):

$$\rho(x_m, y_m, z_m) = \sum_{A=1}^{N} \rho_A \qquad (3)$$

$$\rho_A = \sum_{i=1}^{n_A} a_{A_{isp}} exp(-b_{A_{isp}} R_A) + \sum_{i=3}^{n_A-1} a_{A_{id}} exp(-b_{A_{id}} R_A) + \sum_{i=4}^{n_A-2} a_{A_{if}} exp(-b_{A_{if}} R_A) \qquad (4)$$

where $N$ is the number of atoms in the molecule, $\rho A$ is the $A$ atomic increment in molecular electron density, $a_{A_{isp}}$, $b_{A_{isp}}$, $a_{A_{id}}$, $b_{A_{id}}$, $a_{A_{if}}$, and $b_{A_{if}}$ are the AlteQ atomic parameters describing the i-th sp-orbital, d-orbital, and the f-orbital of the A atom respectively, $nA$ is the period number of the $A$ atom, $R_A$ is the distance between the $A$ atomic center and point $m$. The units of the AlteQ coefficients are $[b] = 1/\text{Å}$, $[a] = e/\text{Å}3$, and consequently $[\rho] = e/\text{Å}3$.

Thus, the electron density of the outer shell, which plays the most important role in the formation of covalent bonds and intermolecular contacts, can be calculated at each point of the molecular space as follows:

$$\rho_{A(outer)} = a_{A_{nsp}} exp(-b_{A_{nsp}} R_A) + a_{A_{(n-1)d}} exp(-b_{A_{(n-1)d}} R_A) + a_{A_{(n-2)f}} exp(-b_{A_{(n-2)f}} R_A). \qquad (5)$$

The AlteQ method separates the molecular density into atomic contributions and allows one to separately consider contributions of the enzyme and the ligand atoms (Eq. 5). The generated electron density maps were used to construct a linear regression model to establish a correlation between the distance and the electron density overlap between the ligand and receptors atoms using Eqs. (1) and (2). The generated 3D electron density

maps were statistically processed using the scikit-learn (*Pedregosa et al., 2011*) and the SciPy library (*Virtanen et al., 2020*) in Python (v 3.7.6), and the plots were generated using the Matplotlib library (*Hunter, 2007*). The AlteQ software is freely available online *via* the ChemoSophia webpage (http://www.chemosophia.com/).

## Molecular dynamics (MD) simulations

MD simulations for all ligands (WS1, WS4, WS11, and two conformations of the WS7 ligand) were run in complex with the SARS-CoV-2 3CL$^{pro}$ in the previously acquired conformation (PDB: 6LU7). The best docked ligand positions, which were determined using the AlteQ method, were used as the starting points for the MD simulations. The AMBER ff14SB force field (*Maier et al., 2015*) was used to model the protein and the GAFF force field (*Wang et al., 2004*) was used to model the ligand. Other simulation parameters such as periodic boundary conditions, NVT conditions, pressure, temperature were the same as described in our previous article (*Novak et al., 2021b*; *Pathak et al., 2021*). In brief, protein-ligand complexes were solvated in a truncated octahedral box of TIP3P water molecules spanning a 12 Åthick buffer, and Na$^+$ and Cl$^-$ ions were added according to (*Machado & Pantano, 2020*) to achieve a neutral environment with a salt concentration of 0.15 M. Such structures were then submitted for geometry optimization in the AMBER16 program (*Case et al., 2016*) employing periodic boundary conditions in all directions. For the first 1,500 cycles, the complex was restrained and only water molecules were optimized, after which another 2,500 cycles of optimization followed where both water molecules and the complex were unrestrained. Optimized systems were gradually heated from 0 to 310 K and equilibrated during 30 ps using NVT conditions, followed by productive and unconstrained MD simulations of 300 ns employing a time step of 2 fs at constant pressure (1 atm) and temperature (310 K), the latter held constant using Langevin thermostat with a collision frequency of 1 ps$^{-1}$. Bonds involving hydrogen atoms were constrained using the SHAKE algorithm (*Ryckaert, Ciccotti & Berendsen, 1977*), while the long-range electrostatic interactions were calculated employing the Particle Mesh Ewald method (*Darden, York & Pedersen, 1993*). The non-bonded interactions were truncated at 11.0 Å. Analysis of the trajectories was performed using the cpptraj module of AmberTools16 (*Roe & Cheatham, 2013*).

## Binding free energy calculations and decomposition

The binding energies, $\Delta G_{BIND}$, of the simulated complexes were calculated using the MM-GBSA (Molecular Mechanics–Generalized Born Surface Area) and the MM-PBSA (Molecular Mechanics–Poisson–Boltzmann surface area) protocols (*Genheden & Ryde, 2015*; *Hou et al., 2011*), available as a part of AmberTools16 (*Ferenczy, 2015*). $\Delta G_{BIND}$ is calculated from snapshots of MD trajectories (*Ferenczy, 2015*) with an estimated standard error of 1–3 kcal/mol (*Genheden & Ryde, 2015*). $\Delta G_{BIND}$ is calculated in the following manner:

$$\Delta G_{BIND} = < G_{complex} > - < G_{protein} > - < G_{ligand} >$$

where the symbol <> represents the average value over 100 snapshots collected from the last 30 ns part of the corresponding MD trajectories (every 150th frame was taken

for the calculation). The calculated MM-GBSA and MM-PBSA binding free energies were decomposed into specific residue contribution on a per-residue basis according to established procedures. This protocol calculates the contributions to $\Delta G_{BIND}$ arising from each amino acid side chains and identifies the nature of the energy change in terms of interaction and solvation energies (*Gohlke, Kiel & Case, 2003*; *Rastelli et al., 2010*). The entropy term was not calculated.

## RESULTS

### Collection of *Withania somnifera* compounds from literature and molecular fingerprinting

A search of the NCBI PubMed database resulted in 1,401 research articles which included the term "*withania somnifera*", with the dates until February 28, 2021. All the articles were reviewed and withanolides were collected. Altogether, a total of 80 compounds were collected. However, several studies have already reported WS withanolides against the SARS-CoV-2 3CL[pro] andhence these compounds were excluded from the analysis. A total of 11 specific types of withanolides, such as withaphysalins, withasomniferols, and withafastuosins, were considered for a detailed binding interaction since there were no previous studies focused on their binding interactions with the SARS-CoV-2 3CL[pro]. Details about withanolides selected in the present investigation and their respective article information are provided in File S1. Among the selected withanolides, withasomniferol A, withasomniferol B, and withasomniferol C are newly isolated WS phytoconstituents (*Anjaneyulu & Rao, 1997*).

Similarity of the collected phytocompounds with reported SARS-CoV-2 inhibitors was checked using a molecular similarity analysis. Tc scores of the phytocompounds were calculated relative to the reported SARS-CoV-2 inhibitors, and they ranged from zero to 0.4 (File S2).

### Molecular docking

From all tested phytocompounds, compounds with ID's WS1, WS4, WS7, and WS11 showed the lowest energies of −8.0, −8.2, −7.6, and −7.8 kcal/mol, respectively, when binding to the SARS-CoV-2 3CL[pro] (File S3). Their molecular interactions with the SARS-CoV-2 3CL[pro] active pocket residues are shown in Figs. 1 and 2, and 2D structures of phytocompounds are shown in Fig. 3. Their binding energies against the 3CL[pro] and their ADME properties are shown in Table 1. The analysis of the ADME properties of potential drug candidates is essential in the early stage of drug discovery to reduce failure rates in the clinical phase of drug discovery. According to Lipinski's rule of 5 (RO5), to be orally active, drug-like compounds should have molecular weight below 500, number of hydrogen bond donors below 5, number of hydrogen bond acceptors below 10, and the log *P* value should not exceed 5. As shown in Table 1, all the selected compounds obey these rules except for the WS1 compound, which has a molecular weight above 500. All four compounds were found to be non-inhibitors of the cytochrome P450 enzymes (CYP3A4, CYP2D6, CYP2C9, CYP2C19, and CYP1A2). The cytochrome P450 enzymes play an essential role in metabolism of various molecules. Their inhibition can cause serious
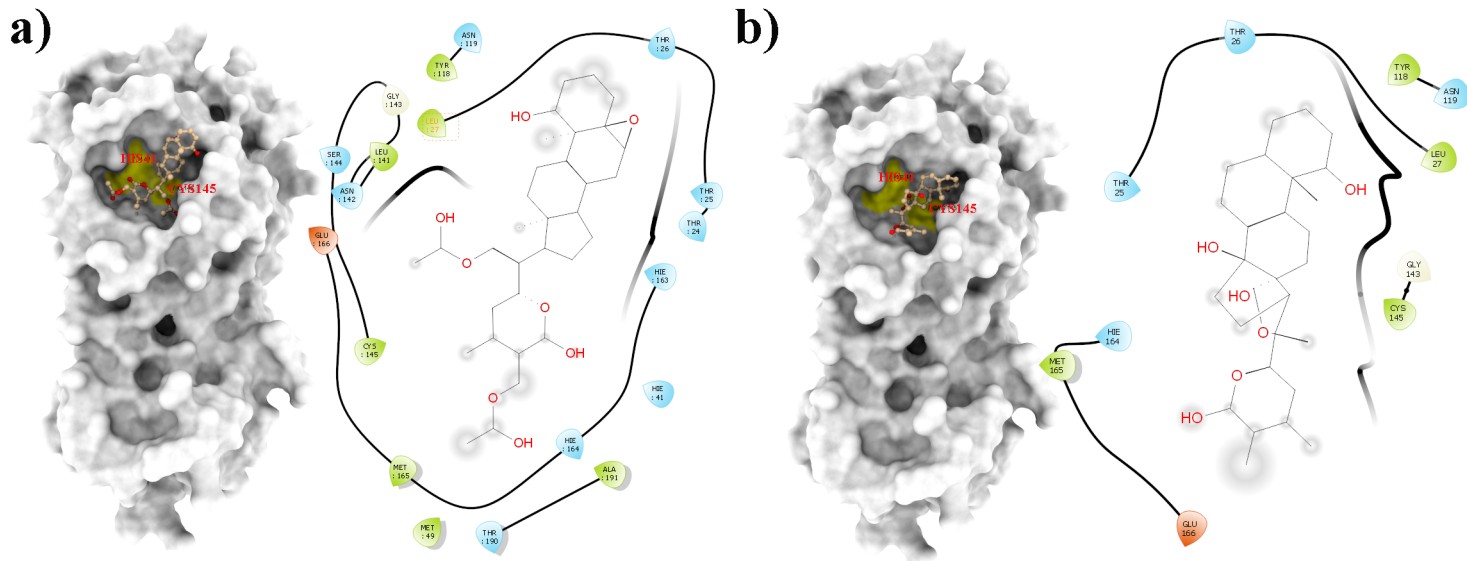

**Figure 1** Molecular interactions of the selected *Withania somnifera* compounds with the Cys145 –His41 catalytic dyad of the SARS-CoV-2 3CL^pro (depicted in yellow) with: (A) WS1 and (B) WS4 (amino acids are depicted in different colors: green: hydrophobic, blue: polar, orange: positively charged, purple: negatively charged, pink: hydrogen bonds).

drug-drug interactions that can cause unanticipated adverse effects (*Lynch & Neff, 2007*). Since all our lead compounds were predicted to be non-inhibitors of the cytochrome P450 enzymes, this suggests that they possess a low potential for drug-drug interactions. Additionally, none of the suggested lead compounds are Pan-assay interference compounds (PAINS), which indicates that they will not give false positive results in high-throughput screens. All PAINS interact nonspecifically with numerous biological targets, as opposed to with specific targets. Furthermore, stability of these compounds with SARS-CoV-2 3CL^pro was verified through MD simulations. Additionally, before the MD analysis, poses of the compounds generated by docking were evaluated through the complementarity centered AlteQ method.

## Selection of the docked poses with the complementarity-centered AlteQ method

Firstly, a linear regression model was constructed to establish a correlation between the distance between the ligand and receptors atoms and their electron density overlap (Eq. 1) for the experimental SARS-CoV-2 3CL^pro complexes and this information was used for evaluating the correctness of the docked conformations. The correlation coefficients ($R^2$) in the 5R7Y, 6M2N, 5RGH, 5RF7, 5RG1, 5RH8, and 5REB complexes were found to be below 0.50 and hence were not included in model generation. The average $R^2$ of all the other experimental conformations was 0.732 (File S4). In a recent article, we described two ways of using the complementary based AlteQ method for validating docking scores (*Rimac, Grishina & Potemkin, 2021*). In the present work, we evaluated the correctness of the docked poses from the intercept-versus-slope graph. The intercept and the slope

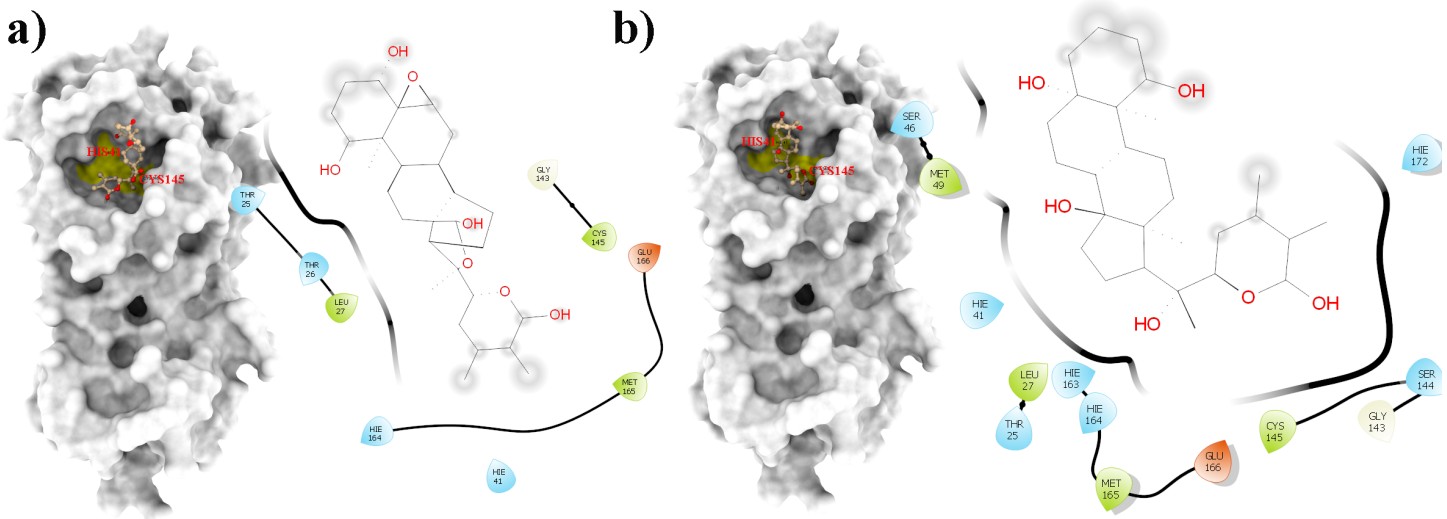

**Figure 2** Molecular interactions of the selected *Withania somnifera* compounds with the Cys145 –His41 catalytic dyad of the SARS-CoV-2 3CL$^{pro}$ (depicted in yellow) with: (A) WS7 and (B) WS11 (amino acids are depicted in different colors: green: hydrophobic, blue: polar, orange: positively charged, purple: negatively charged, pink: hydrogen bonds).

**WS1**

**WS4**

**WS7**

**WS11**

**Figure 3** 2D structures of compounds. (WS1) withafastuosin D, (WS4) withaphysalin D, (WS7) withaphysalin N, and (WS11) withasomniferol C.

coefficients for all experimental complexes collected from Eq. 1 are correlated and fall onto the same regression line ($R^2 = 0.97$). The following model (Eq. 6) was obtained (Fig. 4).

$$a = -2.90 - 0.33 \times b (R^2 = 0.97) \tag{6}$$

where, $a$ is the intercept value and $b$ is the slope value.

**Table 1 Binding energy and ADME properties of withanolide compounds against SARS-CoV-2 3CL^pro.**

| Molecule ID (name) | B.E[a] | MW[b] | RB[c] | RO5[d] | ADME[e] | PAINS[f] |
|---|---|---|---|---|---|---|
| WS1 (withafastuosin D) | −8 | 554.67 | 8 | Yes (MW) | No | No |
| WS4 (withaphysalin D) | −8.2 | 466.57 | 1 | No | No | No |
| WS7 (withaphysalin N) | −7.6 | 484.58 | 1 | No | No | No |
| WS11 (withasomniferol C) | −7.8 | 470.6 | 2 | No | No | No |

**Notes.**
[a] B.E, binding energy (kcal/mol).
[b] MW, molecular weight (g/mol).
[c] RB, number of rotatable bonds.
[d] RO5, Lipinski rule of 5 (violations in the molecular weight is allowed in the analysis).
[e] ADME, important pharmacokinetics properties *i.e.*, cytochrome P450 inhibitors (CYP3A4, CYP2D6, CYP2C9, CYP2C19, and CYP1A2) and blood–brain barrier (BBB) penetration, which were calculated using SWISS ADME (*Cole et al., 2005*).
[f] PAIN, PAINS liabilities checked using SWISS ADME.

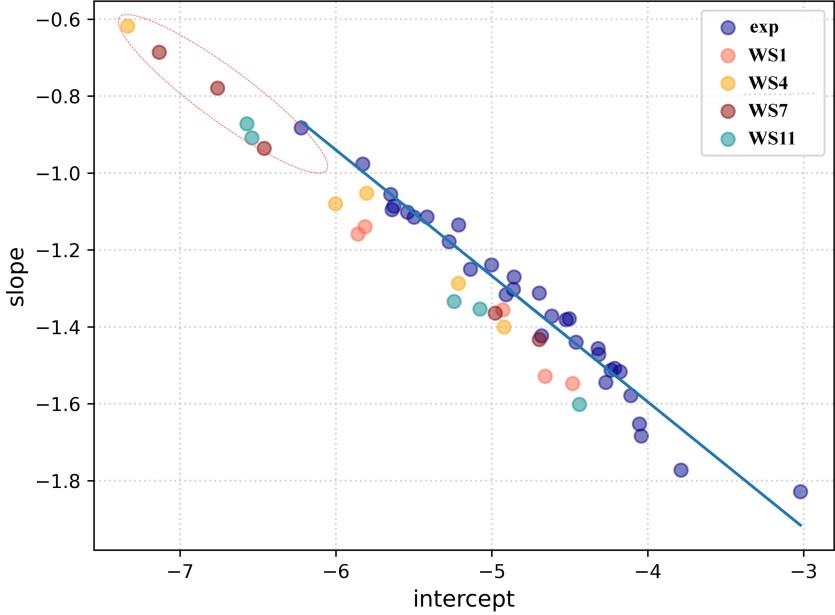

**Figure 4 A linear regression model established by correlating the intercept and the slope (Eq. (6)) for the experimental conformations (dark blue) and the docked conformations (WS1 conformations are shown in light red, WS4 in orange, WS7 in dark red, and WS11 in dark cyan).** Root-mean-square deviation of all non-hydrogen atoms for 3CL^pro complexes with ligands WS1 (red), WS4 (blue), WS7_v1 (cyan), WS7_v2 (olive), WS11 (black).

The distance between the receptor and the ligand decreases with the slope, as can be observed in Fig. 4. In the 5RH1 complex, the ligand and the receptor atoms are very close, and the slope value was found to be −1.83. In contrast, the slope value of the 5R81 complex was found to be very high (−0.88), which indicates long distances between the ligand
and the receptor atoms. The regression line obtained from the experimental SARS-CoV-2 3CL$^{pro}$ complexes was used a reference line to measure appropriateness of the generated docked conformations.

The top five docking poses of WS1, WS4, WS7, and WS11 were evaluated, as they show the lowest binding energies when binding to the 3CL$^{pro}$. Their slope and intercept values were collected from Eq. 1 and plotted against the experimental conformations (Fig. 4). As expected, different electron density overlap patterns were observed in the docked conformations; they are slightly away from the regression line of the experimental conformations due to different conformations and their orientation within the active pocket, and they correspond to different interaction patterns. The distance of the docked conformations from the regression line was measured and the complete information is provided in File S5. In the top five conformations, the distance of the conformation 1 (based on the docking results) of WS1(0.062), WS4 (0.045), WS11(0.105) ligands were found to be much lower compared to the other four conformations, which is in accordance with the experimental conformations. The only exception was conformation 1 of the WS7 ligand, whose distance was found to be much greater (0.111). The electron density overlap patterns of a few conformations of the ligands WS4, WS7, and WS11 were more distant from the rest of the points (Fig. 4, dotted circle), and these conformations show a unique pattern in the electron density overlap, which was not observed in the experimental conformations.

## Molecular dynamics simulations and free energy calculations

To check for the stability of the conformations, MD simulations were performed. For ligands WS1, WS4, and WS11 the best docked conformations were chosen since their electron density overlap patterns were very similar to the electron density overlaps of the experimental conformations. For the WS7 ligand, two different conformations were considered, namely the first (−7.6 kcal/mol, WS7_v1) and the fifth conformation (−7.3 kcal/mol, WS7_v2), with both conformations having different electron density overlap patterns with different binding energies. The distance of the conformation 1 of the WS7 ligand (0.111) was found to be much higher when compared to the fifth conformation (0.062) from the regression line, but the fifth conformation showed a unique pattern in the electron density overlap, and its points (*i.e.*, slope and intercept values from Eq. 3) are located far away from the rest of the points (Fig. 4, dotted circle).

From the five complexes tested, three complexes were found to be stable for the entire duration of the simulation (WS4, WS7_v1, and WS11), while two complexes showed a brief dissociation (∼8.5 ns in the case of WS7_v2 and ≈ 26.0 ns in the case of WS1), which can also be seen from Fig. 5. The root-mean-square fluctuation (RMSF), radius of gyration and intermolecular hydrogen bond plots are provided in the File S6.

From Table 2, it can also be seen that WS1 and WS7_v2 bind the weakest (calculated using the MM-GBSA approach). From the three stable compounds (WS4, WS7_v1, and WS11), WS7_v1 and WS11 have approximately the same $\Delta G_{BIND}$, which is significantly lower than that of WS4. Additional calculations of $\Delta G_{BIND}$ were performed using the MM-PBSA protocol. These results are completely in accordance with the MM-GBSA results and show the same trend; WS7_v1 and WS11 have virtually the same binding
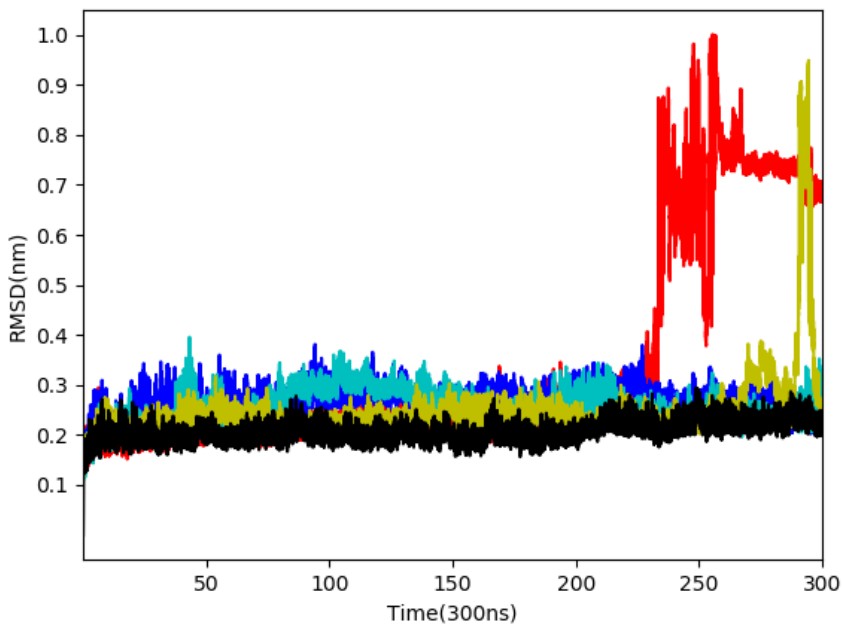

**Figure 5** **Root-mean-square deviation of all non-hydrogen atoms for 3CL^pro complexes with ligands WS24 (red), WS51 (blue), WS54_v1 (cyan), WS54_v2 (olive), WS81 (black).**

constant ($\sim$−26.00 kcal/mol), with the other three ligands having at least 12 kcal/mol higher $\Delta G_{BIND}$ (detailed information can be found in File S7). The top ten contributing amino acids for the binding of the two best ligands (*i.e.*, WS7_v1 and WS11) are shown in Table 3 and their positions relative to the catalytic Cys145 residue are shown in Fig. 6. Relative to the figure, the WS7_v1 ligand is located slightly more to the left, and the WS11 ligand slightly more to the right. This results in a fact that these ligands share nine out of ten most contributing residues (Table 3), with exception being Thr45, which interacts only with the WS7_v1 far left hydroxy group, and Met165, which interacts only with the WS11 cyclopentanyl group (Fig. 7, Table 3, bolded). While the hydrogen bond interaction with Thr45 is moderately important in case of WS7_v1 binding, the van der Waals interaction with Met165 seems to be the most important one for WS11_v1 binding. Additionally, the WS7_v1 ligand forms two very strong interactions with partial $\Delta G_{BIND}$ lower than −2.0 kcal/mol, with one of them being the Cys145 residue, a crucial residue for the 3CL^pro function (*Cole et al., 2005*), and the other one being Thr25 which forms a very strong hydrogen bond with the keto oxygen atom of WS7. On the other hand, WS11 forms much stronger interactions with the other crucial residue, (His41), as well as with Cys44, with both interactions being hydrophobic in nature, while in the case of WS7_v1 these interactions have a van der Waals character. This indicates that WS7 and WS11 interactions have a preference for different parts of the binding pocket and that they could both be good candidates for 3CL^pro inhibitors. This also implies that there is still room for developing or finding a ligand which could interact more strongly with both parts of the binding pocket

**Table 2** **Average number of intermolecular hydrogen bonds and $\Delta G_{BIND}$ (MM-GBSA) of withanolide compounds against SARS-CoV-2 3CL$^{pro}$.**

|  | H-bonds | | $\Delta G_{BIND}$[a] |
|---|---|---|---|
|  | mean | s.d. | (kcal/mol) |
| WS1 | 0.90 | 1.02 | −17.20 |
| WS4 | 1.01 | 0.89 | −24.16 |
| **WS7_v1** | **2.45** | **1.47** | **−35.54** |
| WS7_v2 | 1.10 | 1.05 | −8.52 |
| **WS11** | **1.87** | **1.33** | **−32.19** |

**Notes.**
[a] Last 30 ns of the 300 ns simulation using MM-GBSA.
Bolded values indicate the compounds that showed the lower deltaG$BIND$.

**Table 3** **The top ten contributing amino acid residues for binding of withanolide compounds WS7_v1 and WS11 to 3CL$^{pro}$. $\Delta G_{BIND}$ values are given in kcal/mol.**

| WS7_v1 | | WS11 | |
|---|---|---|---|
| Residue | $\Delta G_{BIND}$ | Residue | $\Delta G_{BIND}$ |
| Thr 25 | −2.55 | **Met 165** | −1.82 |
| **Cys 145** | −2.27 | Cys 44 | −1.68 |
| Gly 143 | −1.46 | **Cys 145** | −1.48 |
| Ser 144 | −1.11 | **His 41** | −1.24 |
| Leu 27 | −1.08 | Gly 143 | −1.11 |
| Cys 44 | −1.03 | Met 49 | −1.03 |
| **Thr 45** | −1.02 | Leu 27 | −1.02 |
| Asn 142 | −0.98 | Thr 25 | −0.82 |
| **His 41** | −0.84 | Ser 144 | −0.81 |
| Met 49 | −0.77 | Asn 142 | −0.66 |

**Notes.**
Bolded values indicate the important residues of 3CLpro.

at the same time. The complete binding free energy results of all the ligands are provided in the File S7.

# DISCUSSION

Natural products play a significant role in the discovery of novel and effective therapeutics to fight the present COVID-19 pandemic. Herbal extracts and spices are natural immune boosters and/or anti-infective agents currently used in many parts of the world (*Gbadamosi, 2020*). In traditional folk medicine, spices, botanical detoxifiers, antioxidants (*Gbadamosi & Afolayan, 2016*) and plant hematinics (*Gbadamosi, 2012*) are used as antiviral mediators to prevent or minimize the impact of various diseases. A lack of targeted treatments has encouraged the exploration of novel drug lead compounds in which computational approaches offer a comparatively fast and cost-effective approach. COVID-19 is characterized by a disrupted immunological balance, hyperinflammation, cytokine storm, and multiorgan failure (*Saggam et al., 2021*). WS plant is reported to mitigate early disease progression and protect vital organs (*Saggam et al., 2021*). In addition, *Saggam et al.*

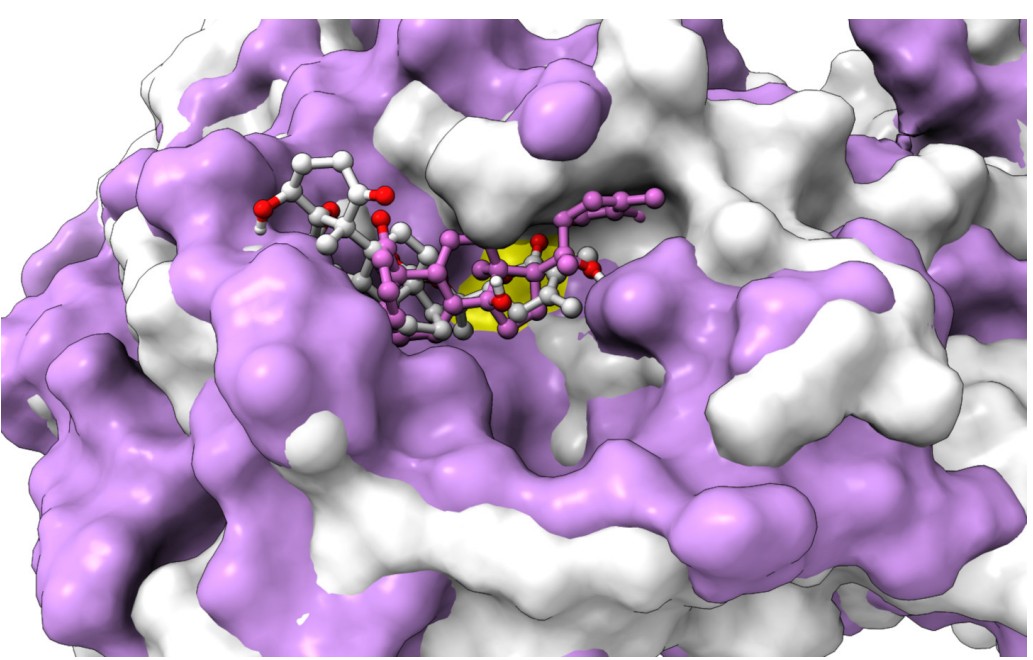

**Figure 6** An overlay of the 3CL^pro –WS7_v1 (white) complex and the 3CL^pro –WS11 (purple) complex active site after 300 ns MD simulations with the Cys145 amino acid residue surface depicted in yellow.

*(2021)* suggested that the WS plant's antiviral capabilities could disrupt viral entrance and its life cycle. The WS phytocompounds were explored for their antiviral potential against SARS-CoV-2 proteins using a computational molecular docking tools. Also, recent reports established that the phytocompound withanone disrupts the host–virus interaction by destabilizing the ACE2–spike protein receptor-binding domain complex (*Balkrishna et al., 2021*). Withaferin A and withanone were proven to prevent viral entry and replication by blocking the 3CL^pro and TMPRSS2 enzymes (*Kumar et al., 2022*). *Kushwaha et al. (2021)* tested non-characteristic phytocompounds (quercetin-3-rutinoside-7-glucoside, rutin, and isochlorogenic acid B) present in the WS plant against SARS-Cov-2 3CL^pro through molecular docking studies. All the above reports focused and discussed computational approaches and further concluded the effect of WS phytocompounds in blocking the viral entry and replication. However, the reports of WS compounds against SARS-CoV-2 are lacking and the amount of experimental and clinical data is limited. This highlights the necessity of performing systematic research in COVID-19 to investigate WS phytocompounds as antiviral therapeutics. According to *Lurie, Keusch & Dzau (2021)*, given recent indications that the SARS-CoV-2 pandemic will be a long-term health problem, there will be a considerable demand for SARS-CoV-2 medicines and adjuvants research and development.

Therefore, in the present study, a total of 11 reviewed withanolides from WS, belonging to the group of withaphysalins, withasomniferols, and withafastuosins, were collected through manual literature curation. The initial molecular docking analysis revealed that the binding energy of all the selected phytocompounds against the SARS-CoV-2 3CL^pro

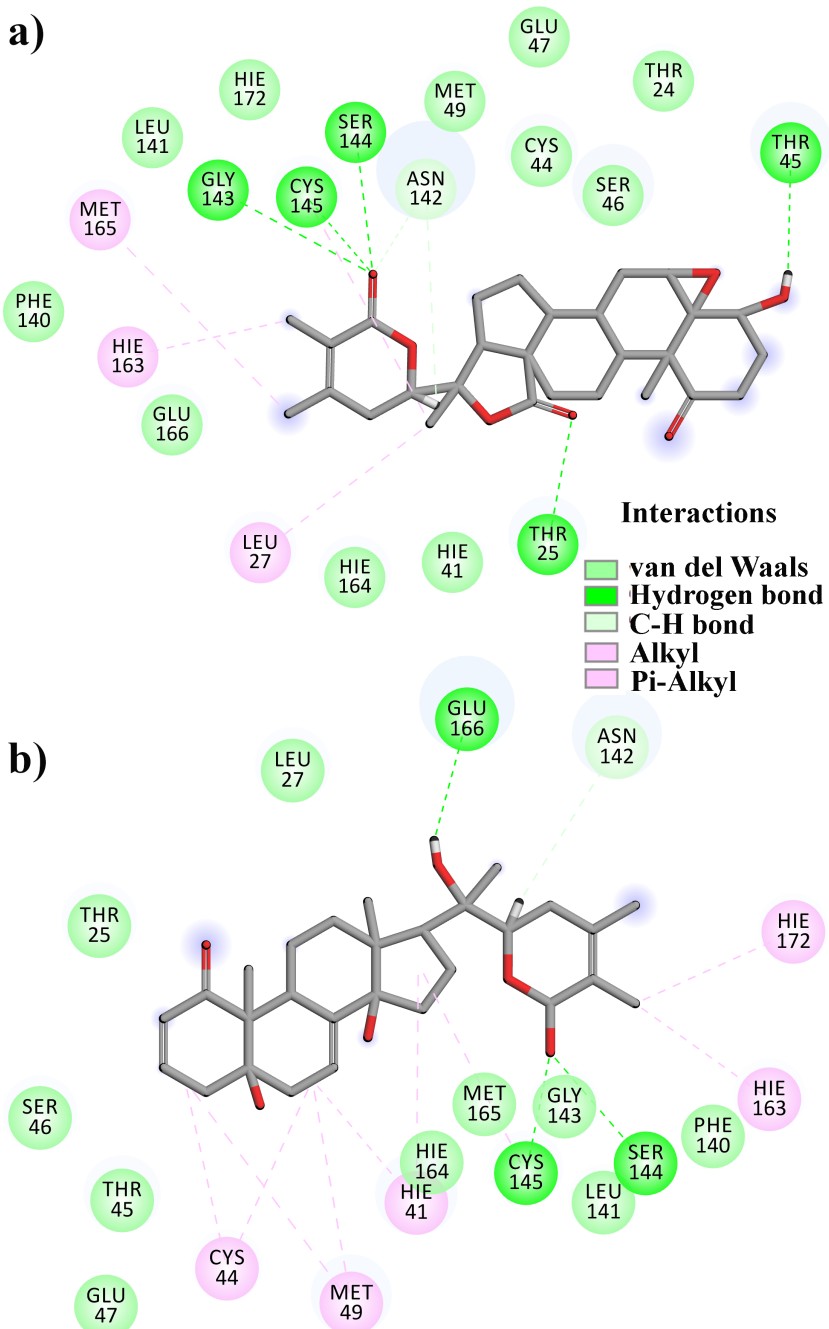

**Figure 7** A 2D representation of withanolides and 3CL^pro important interactions. (A) WS7_v1 (B) WS11 (interactions are depicted in different colors: lawn green—conventional hydrogen bonds, light green—van der Waals interactions, tea green—carbon hydrogen bond, pink—alkyl and pi-alkyl interactions).

ranged between –6.3 and –8.2 kcal/mol. Among them, WS4, WS1, WS11, and WS7 had the lowest binding energies to the SARS-CoV-2 3CL$^{pro}$. Validation of the obtained docked conformations were assessed using the complementarity principle implemented in the AlteQ method. It is a recently introduced method for determining ligand–receptor contacts in which no cut-offs for the length of ligand–receptor contacts are included (*Potemkin & Grishina, 2008*). The constructed linear regression model of the experimental SARS-CoV-2 3CL$^{pro}$ complexes was found to be reliable, and the average correlation coefficient was found to be 0.732. The obtained regression line of the experimental SARS-CoV-2 3CL$^{pro}$ complexes was used as a reference line to measure the appropriateness of the generated docked conformations. The appropriateness of the docked conformations generated by the docking tool was measured based on their distance from the obtained regression line of the experimental SARS-CoV-2 3CL$^{pro}$ complexes. The distance of the conformation 1 of the WS1, WS4, and WS11 compounds was found to be very low compared to the other conformations from the regression line of the experimental conformations. However, for the WS7 compound, the distance of the conformation 5 was found to be the lowest. This indicates that these docked conformations have similar intermolecular electron density overlaps as the experimental SARS-CoV-2 3CL$^{pro}$ conformations. Further validation of these conformations in MD studies showed that all conformations except WS1 show a stable interaction with the SARS-CoV-2 3CL$^{pro}$. The WS1 conformation showed a brief dissociation (∼26.0 ns) during the MD simulation. It is interesting to note that docking of small ligands with six or fewer rotatable bonds is very fast and accurate (*Plewczynski et al., 2011*) and all compounds except WS1 have fewer than three rotatable bonds. Hence, it can be concluded that the docking procedure generated correct poses for the tested compounds. Finally, binding free energy calculations and the decomposition analysis showed a high binding affinity of WS7_v1 (withaphysalin D) and WS11 (withasomniferol C). These ligands interact with the key 3CL$^{pro}$ residues, including a strong interaction with the Cys145 –His41 catalytic dyad (Table 3), which is crucial for the 3CL$^{pro}$ function. The conformation 5 of WS7 (v2) ligand failed to show a low binding free energy because of its unique electron density pattern (Fig. 4, dotted circle), which was not observed in the experimental conformations.

Several docking studies reported binding affinities of WS plant phytocompounds against the SARS-CoV-2 3CL$^{pro}$. On the other hand, we calculated binding affinities from molecular dynamic simulations because as it is more reliable than docking binding energy prediction (*Pagadala, Syed & Tuszynski, 2017*; *Pantsar & Poso, 2018*). Previous reports that reported binding of WS phytocompounds to the SARS-CoV-2 3CLpro had a few drawbacks, *i.e.*, (1) binding free energy calculations for some phytocompounds were not performed after MD simulations, (2) the simulation time of most of the previously reported phytocompounds was less than 100 ns, (3) additional validation of binding free energy calculations was not performed. However, in the present investigation, the duration of MD simulations was 300 ns, after which the binding free energy was calculated with both the MM-GBSA and the MM-PBSA protocols. The binding free energies of the reported phytocompounds from the WS plant against the SARS-CoV-2 3CL$^{pro}$ are provided in the File S8. It is important to say that a compound presented in this study, namely WS11 (withasomniferol C), showed

the lowest binding free energy against the SARS-CoV-2 3CL$^{\text{pro}}$ compared to all previously reported WS phytocompounds.

## CONCLUSIONS

Altogether, the present investigation gives a comprehensive overview of the specific types of withanolides and shows their potential against SARS-CoV-2 3CL$^{\text{pro}}$. The *in silico* analysis elucidated two potential candidates, namely WS7 (withaphysalin D) and WS11 (withasomniferol C) as potential 3CL$^{\text{pro}}$ inhibitors. Among them, withasomniferol C is newly isolated WS phytoconstituents and showed the lowest binding free energy against SARS-CoV-2 3CL$^{\text{pro}}$. Nevertheless, additional *in vitro* and *in vivo* studies are essential to validate their efficacy as good drug candidates for inhibition of the 3CL$^{\text{pro}}$ activity.

## ACKNOWLEDGEMENTS

The authors also acknowledge University of Zagreb, University Computing Centre (SRCE) for granting computational time on the Isabella cluster.

### Funding
This study was funded by RFBR, DST, CNPq, SAMRC, project number 20-53-80002. The funders had no role in study design, data collection and analysis, decision to publish, or preparation of the manuscript.

### Grant Disclosures
The following grant information was disclosed by the authors:
RFBR, DST, CNPq, SAMRC: 20-53-80002.

### Competing Interests
The authors declare there are no competing interests.

### Author Contributions

- Shivananada Kandagalla conceived and designed the experiments, performed the experiments, analyzed the data, prepared figures and/or tables, authored or reviewed drafts of the article, and approved the final draft.
- Hrvoje Rimac performed the experiments, analyzed the data, prepared figures and/or tables, authored or reviewed drafts of the article, and approved the final draft.
- Krishnamoorthy Gurushankar conceived and designed the experiments, performed the experiments, analyzed the data, authored or reviewed drafts of the article, and approved the final draft.
- Jurica Novak performed the experiments, analyzed the data, prepared figures and/or tables, authored or reviewed drafts of the article, and approved the final draft.
- Maria Grishina performed the experiments, analyzed the data, authored or reviewed drafts of the article, and approved the final draft.

- Vladimir Potemkin performed the experiments, analyzed the data, authored or reviewed drafts of the article, and approved the final draft.

## Data Availability

The raw data are available in the Supplementary Files.

## Supplemental Information

Supplemental information for this article can be found online at http://dx.doi.org/10.7717/peerj.13374#supplemental-information.

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
