# Peer review of "Withasomniferol C, a new potential SARS-CoV-2 main protease inhibitor from the Withania somnifera plant proposed by in silico approaches"

_PeerJ, doi:10.7717/peerj.13374_

## Round 0.1 · original submission · Major Revisions

Note that your manuscript has to include more details about the methodology, to consider papers dealing with the same subject, to have the language corrected by someone fluent in English and the originality has to be emphasized since several papers have already appeared about the effects of the studied plant.

Reviewer 1 ·

Basic reporting

1- pg 5, line 25. In the word "com-pounds" remove the hyphen.
2- pg 6, line 39. Please, remove the hyphen from the word "pa-tients".
3- pg 7, line 82. The word "in silico" must to be in italics.
4- pg 7, line 105. The word "Solanaceae" must to be in italics.
5- pg 7, line 115. Why did the authors cited the "NSP15"? Please, check if it is correct.
6- pg 9, line 167. Please, remove the hyphen from the word "hydro-gens".
7- pg 10, lines 202 and 208. In the word "mth", need the "th" to be overwritten?
8- pg 12, line 281. In the word "150th", need the "th" to be overwritten? Please, check it in all text.
9- pg 12, line 282. It is missing a space between the words "MM-PBSAbinding".
10- pg 12, line 302. Please, remove the hyphen from the word "phytocom-pounds". And check it in all text to find other words with typos.

Experimental design

1- pg 8, lines 154-155. Please, indicate the force field used to optimize the molecules.
2- pg 12, lines 313-314. Please, add a comment about the docking results from the cited molecules. Is the data founded by the authors similar to the literature?
3- pg 27. Figure 2. Please, improve the quality of the figure 2. It is not possible to read the amino acids residues names and the interactions legend.
4- pg13, lines 332-334. Please, explain better why molecules with a poor BBB penetration are good drug candidates? Taking into account the neuroinvasive ability of SARS-CoV-2 (doi 10.1084/jem.20202135), please, comment the data.

Validity of the findings

no comment

Additional comments

no comment

Reviewer 2 ·

Basic reporting

1.1. Please revise the writing of the entire manuscript before resubmitting it.

1.2. In the abstract, name the compounds WS54 (withaphysalin D) and WS81 (withasomniferol C).

1.3. Some unnecessary hyphens in words are found, e.g.: com-pounds (line 25), in-volved (line 76), hydro-gens (line 167), con-formation (line 183), re-ported (line 424), and so on.

1.4. Different paragraphs are concentred in a unique paragraph in lines 97-121, making it too long. Please revise it.

1.5. "According to the Lurie et al., 2021" (line 421), change to "According to Lurie et al. (2021)";
"Saggam et al., 2021 suggested" (line 425), change to "Saggam et al. (2021) suggested"; "Kushwaha et al., 2021," (line 432) change to "Kushwaha et al. (2021), and so on.

1.6. The figures must be improved. Revise the resolution and the size of words and number of graphs in figures (they are too short to visualize, mainly figures 2, 5, and 7), according to the dimensions of the manuscript page.

1.7. The captions must be improved and self-explanatory to the best comprehension (e.g. see figure 2).

1.8. The 2D structures of compounds from Table 1 are too short. Please look for a better way to show these structures in the manuscript.

1.9. Improve the title of table 2 (Where these ΔGBIND were calculated? What methodology was employed?).

1.10. Titles from figures and tables must be objective but self-explanatory.

1.11. To meet the standard spelling of amino acid residue and number position in the primary protein sequence, avoid space between amino acid code and number position. Write "Thr25" instead "Thr 25", Cys145 instead "Cys 145", and so on. Revise the entire manuscript, figures, and tables.

Experimental design

2.1. In the "Molecular docking" section, detailed information about the construction of 3D structures of ligands in detail is missed. How were the geometry of the structures optimized? Quantum (semiempirical or DFT) methods, classical method (force field)? What are the procedures adopted by the authors concerned with the accuracy of 3D structures with global minima of energy? Accurate ligand structures are essential for the subsequent docking and MD studies.

2.2. Despite the information is presented in tables and figure 7, the description of the interactions shown in lines 403-409 must be improved and detailed between each compound and residues.

Validity of the findings

3.1. Despite the interactions of WS69 (somniferine), WS42 (withanoside II), WS34 (withanolide J), WS33 (withanolide F), and WS75 (withanolide L) with the 3CLpro were already reported previously, it is important to compare the inhibitory activity of these compounds with the two new promissory compounds presented by this study WS54 (withaphysalin D) and WS81 (withasomniferol C). Are they present the same potency or more inhibitory activity?

3.2. The authors also excluded common phenolic, organic acids, and other natural compounds from the analysis with known activity against SARS-CoV-2 3CLpro in vitro (lines 443-450). Have some molecular modeling studies of these excluded compounds been reported? Explain this better in the manuscript. If no, present a good justification to explain why they were excluded.

3.3. It is important to compare these results with previous in vitro data (e.g. IC50 values, Km or others from isolated compounds or W. somnifera extract), to infer the involvement of the compounds in the activity.

Additional comments

In the introduction section (lines 117-121), the authors highlighted that "no studies have been performed to test all the reported WS phytoconstituents against SARS-CoV-2. In this research, all the reported WS phytoconstituents were collected through manual literature search, and their binding efficiency against SARS-CoV-2 3CLpro was evaluated using in silico molecular docking and MD simulation studies." In the way it is set out, the understanding is that all the 81 compounds were submitted to molecular docking and MD simulation, but that is not so. Compounds were excluded from the analysis (see comments here in section 3 "Validity of the findings"). The authors must revise and correct this fragment.

A set of reports concerning the inhibition of SARS-CoV-2 main protease by compounds from Withania somnifera has been recently published:
Tripathi, M. K., Singh, P., Sharma, S., Singh, T. P., Ethayathulla, A. S., & Kaur, P. (2021). Identification of bioactive molecule from Withania somnifera (Ashwagandha) as SARS-CoV-2 main protease inhibitor. Journal of Biomolecular Structure and Dynamics, 39(15), 5668-5681.
Dhawan, M., Parmar, M., Sharun, K., Tiwari, R., Bilal, M., & Dhama, K. (2021). Medicinal and therapeutic potential of withanolides from Withania somnifera against COVID-19. J. Appl. Pharm. Sci, 11, 6-13.
Sudeep, H. V., Gouthamchandra, K., & Shyamprasad, K. (2020). Molecular docking analysis of Withaferin A from Withania somnifera with the Glucose regulated protein 78 (GRP78) receptor and the SARS-CoV-2 main protease. Bioinformation, 16(5), 411.
Khanal, P., Chikhale, R., Dey, Y. N., Pasha, I., Chand, S., Gurav, N., ... & Gurav, S. (2020). Withanolides from Withania somnifera as an immunity booster and their therapeutic options against COVID-19. Journal of Biomolecular Structure and Dynamics, 1-14.
Kushwaha, P. P., Singh, A. K., Prajapati, K. S., Shuaib, M., Gupta, S., & Kumar, S. (2021). Phytochemicals present in Indian ginseng possess potential to inhibit SARS-CoV-2 virulence: A molecular docking and MD simulation study. Microbial pathogenesis, 104954.

The methodology employed and the results obtained are adequate and relevant. However, considering the works previously published, the authors must rethink the title, justification, and objective of the work, focusing on the novelty and the relevance of the data to the readers of PeerJ.

---

## Round 0.2 · Minor Revisions

The reviewers have recommended the publication of your manuscript, but the figures need to be improved. The title has to be also corrected and has to be more conservative. You have no in vitro experimental evidence of inhibition, so state that the drug is a potential or putative inhibitor of the enzyme.

Reviewer 1 ·

Basic reporting

The authors made the corrections from the first version of the manuscript, according to the reviewers comments, improving the quality of the study.

Experimental design

The authors made the corrections from the first version of the manuscript, according to the reviewers comments, improving the quality of the study.

Validity of the findings

The authors made the corrections from the first version of the manuscript, according to the reviewers comments, improving the quality of the study.

Additional comments

Given the data presented, the title can be modified to "Withasomniferol C, a new potential SARS-CoV-2 main protease inhibitor from the Withania somnifera plant identified by in silico studies".

Reviewer 2 ·

Basic reporting

The title must be improved to: Withasomniferol C, a new SARS-CoV-2 main protease inhibitor from the Withania somnifera plant proposed by in silico approaches.

line 162: Improve the sentence to (dimensions: x = 13.3, y = 58.2, z = 45.4).

Figure 1: The resolution of the figure is now satisfactory, but the font size of captions highlighted amino acids of the active binding site still remains too short to read in the figure 1. You must edit the image in an other image editor program to increase the font size to improve the visualization to readers.

Figures 5 and 7: the font size of information from the y-axis and x-axis in figure 5; and captions of interactions in figure 7 still remains too short to read in the final version of the manuscript (consider the final manuscript, where the size of figures will be adjusted to the manuscript page, they probably will diminish the figure size). The font size must be also increased.

Experimental design

No comments.

Validity of the findings

No comments.

Additional comments

No comments.

---

## Round 0.3 · Minor Revisions

Thank you for revising the manuscript. However, before accepting I recommend you to change the title to: "Withasomniferol C, a new potential SARS-CoV-2 main protease inhibitor from the Withania somnifera plant proposed by in silico approaches". Furthermore, check and improve the quality of figure 1. The name of the residues are hard to see and the quality of the text (color indications) in the bottom of the figure are also in very poor resolution. Please, also check the indication of His residues as HIE in figure 6.

---

## Round 0.4 · Minor Revisions

Thank you for the corrections, but figure 1 and 2 are still in low resolution. Please, increase the resolution of them.

---

## Round 0.5 · accepted · Accept

Thank you for revising the title and figures 1 and 2.